# The Polychrome in Expression of Baroque Façade Architecture

**Bogna Ludwig**

Faculty of Architecture, Wroclaw University of Science and Technology, 50-317 Wrocław, Poland;
bogna.ludwig@pwr.edu.pl

**Abstract:** The article is dedicated to the role of polychrome solutions of the architectonic order in shaping the mode of expression of Baroque façades. The ancient principles of designing architectural structures, inherited from the Renaissance, were subjected to reinterpretations in order to impart different expressive values. Vertical layouts began to dominate in the Baroque. Appropriately selected polychrome of the elements of the order could emphasize the compositional expression. The relationship between the layout of the polychrome in a given architectural order and the expression of a work of art has been established for some time. However, the generally available data on color schemes of architectural structures in Baroque buildings are still not fully organized. The paper analyzes examples of Baroque façades preserved in their original state and revalorized in recent years after thorough research by conservators in the field of architecture and color. The examples are mainly designed in the so-called great order, i.e., pertaining largely to church façades. The decisive field of change became the shaping of the coloristic decoration of the entablature.

**Keywords:** polychrome; baroque architecture; 16th–18th century; entablature; architectural order

## 1. Introduction

Baroque architecture affects the viewer due to the expression of its composition as shaped by means of carefully selected forms, profiles, connections and proportions of the elements of the order. The ancient principles of designing architectural structures, inherited from the Renaissance and already fully known and refined, were subjected to reinterpretations as early as the late Renaissance and Mannerism in order to impart different expressive values. Vertical layouts began to dominate in the Baroque. From the conceptions of Borromini and Guarini, the primacy of the compositional expression over the maintenance of faithfulness to the rules of architectural order was marked. Thus a developing trend known as the *Aclassical* Baroque exposed the importance of the dynamics of forms and vertical arrangements in façade tectonics.

In the case of the first Italian Baroque realizations, whole façades, or at least architectural details, were made of stone and preserved in their natural color, and only sometimes the background of the façade was painted. However, in the further evolution, especially in the projects of South German architects, appropriately selected polychrome of the elements of the order could emphasize the compositional expression of architectural structures. The decisive field of change became the shaping of the coloristic decoration of the entablature.

## 2. Status of Research: Materials and Methods

For years, the issues of the coloring of Baroque façades have been dealt with by conservators, who study the preserved polychromes and reproduce them in comprehensive restoration projects. This information is collected in the form of reports on the work carried out. Few of the research results are presented in the form of articles (e.g., Koller 1998, 2010; Storemyr 2001). There appear single studies summarizing achievements in selected thematic areas, e.g., concerning a certain group of architectural objects or a geographical region (Knoepfli 1965; Philippot et al. 1986; Grognardi and Tagliasacchi 1988; Brzezowski 2000; Dettloff 2010; Koller 2017). The issue is also discussed along with general problems of

restoring original coloring of historical objects (Muratore 2010). The relationship between the layout of the polychrome in a given architectural order and the expression of a work of art has been established for some time (Zander 1984). However, the generally available data on color schemes of architectural structures in Baroque buildings are still not fully organized.

The paper will analyze examples of Baroque façades preserved in their original state and revalorized in recent years after thorough research in the field of architecture and color by conservators. The examples are mainly designed in the so-called great order, i.e., pertaining largely to church façades. The role of polychrome solutions of individual elements of the order structure in the architectural concept of the Baroque façade will be defined. Describing particular color arrangements will make it possible to point out the most important differences determining the changes of architectural expression. The aim will be to identify the types of solutions and to present the most probable genesis of individual design ideas. The presented methods of selecting color schemes will reveal specific Baroque solutions used to emphasize the tectonics of the building and shape artistic effects with their help.

### 3. Historical and Problematic Background

*Architectural Orders in Modern Times and Forms Expresion*

Redefinition of classical arrangements and saturation of individual details with new solutions and changes in architectural profiles provided an opportunity to enrich expression in the Baroque period. The appropriate selection of elements, their size, plasticity, and profiling, the interrelation of individual elements, as well as the saturation with decoration and arrangement on the façade determined the expressive character of the architectural composition. Small architectural details, slightly protruding from the wall, created a delicate, linear decoration. In this way, the wall took on the form of a sheet, enclosing the space in a strictly defined place, both in direct view and from a perspective. Massive, prominent forms, with a strong chiaroscuro dynamized the architecture, introduced the impression of movement of space, and blurred the spatial boundaries. Entablatures, cornices, and parapets emphasizes horizontal features and made the objects seem lower and optically heavier, binding them to the ground and closing the space. The balance of both divisions created on the façades a kind of network that stabilized the composition. Particular articulation of such elements as columns, pilasters, lisens, risalitic walls, and the corner entablature above the supports gave the buildings a vertical character. This determined their apparent height or slenderness and created the illusion of lightness, sometimes even dematerialization.

Changes in architectural dynamics during the late Renaissance, Mannerism, and early Baroque consisted of a transformation of the balanced layout in favor of bivalent composition, and then the predominance of verticalism. This was largely connected with the breaking of the optical continuity of the entablature, embattled above the supports. The layout with cornices protruding above the columns was already known from Roman architecture, especially from the triumphal arches; the best-preserved ones include Titus, Septimius Severus, and Constantine. It was previously used by Alberti (Tempio Malatestiano Rimini). These compositional solutions became popular through the publications of Serlio and Palladio. In "I quattro libri dell'architettura", Andrea Palladio presented examples of façades composed on the basis of these patterns (Palladio [1570] 1581, L. II, p. 9). The architect himself also used them in projects such as the Teatro Olimpico and the Loggia del Capitanio, as well as the Palazzo Valmarana in Vicenza (1565).

However, the greatest significance for the spread of the triumphal arch composition in monumental architecture had the model of the church façade in the form of the front of the Jesuit church Il Gesù in Rome (Giacomo della Porta, 1573–1580, Figure 1a) (Tafuri 1981). Such a structural solution became almost a standard adopted in the design of all Jesuit church façades but was also used for other religious and diocesan churches. The activity of Northern Italian builders contributed to the spread of such architectural solutions

in other Central European countries, e.g., the Republic of Poland (J.M. Bernardoni; see Appendix A, A1) (Stankiewicz 2016; Betlej 2018). Similar factors were decisive for the widespread use of strongly embattled entablature in mature and late Baroque buildings in southern Germany, Austria, the Bohemia with Silesia, and the Commonwealth of Poland and Lithuania. The builders intensely verticalized the façade composition. This can already be seen in the work of Carlo Antonio Carlone (see Appendix A, A2) (Schmeller 1957). Such increasingly expressive solutions can also be traced in the work of the Dientzenhofer family: Leonhard, for the Jesuit church and Neue Residenz in Bamberg, the façade of the monastery church in Schöntal (1707), and the monastery in Banz; Johann, for the façade of Fulda Cathedral and the monastery church in Banz; and Christoph and Kilian Ignaz, for the façade of the Church of St. John of Nepomuk on the Rock in Prague, St. Nicholas in the Mala Strana, the monastery church in Legnickie Pole, and the parish churches in Hořice and Nepomuk. Analogous compositions were used by late Baroque architects, including Italian artists inclining to Classicism, such as Domenico Rossi on the façade of the church of Santa Maria Assunta in Venice (1715) or Ferdinando Fuga on the façade of S. Maria Maggiore in Rome (1741–1743), as well as artists from Germany and Central Europe, such as Balthasar Neumann, who strove to increase the dynamics of form, for example, in the façade of the Holy Trinity Pilgrimage Church in Goessweinstein (1730–1739) and the Basilica of the Fourteen Holy Helpers and the abbey church in Neresheim (1747–1792). Cornices protruding above supports appeared in almost all works of Paweł Antoni Fontana (1696–1765), who worked in the Commonwealth of Poland; the Brothers of Mercy Church of the Holy Trinity in Cracow (1751–1758), designed by Franciszek Placidi; and the Church of St. Sophia in Polotsk (1738–1750) by Jan Krzysztof Glaubitz from Lithuania.

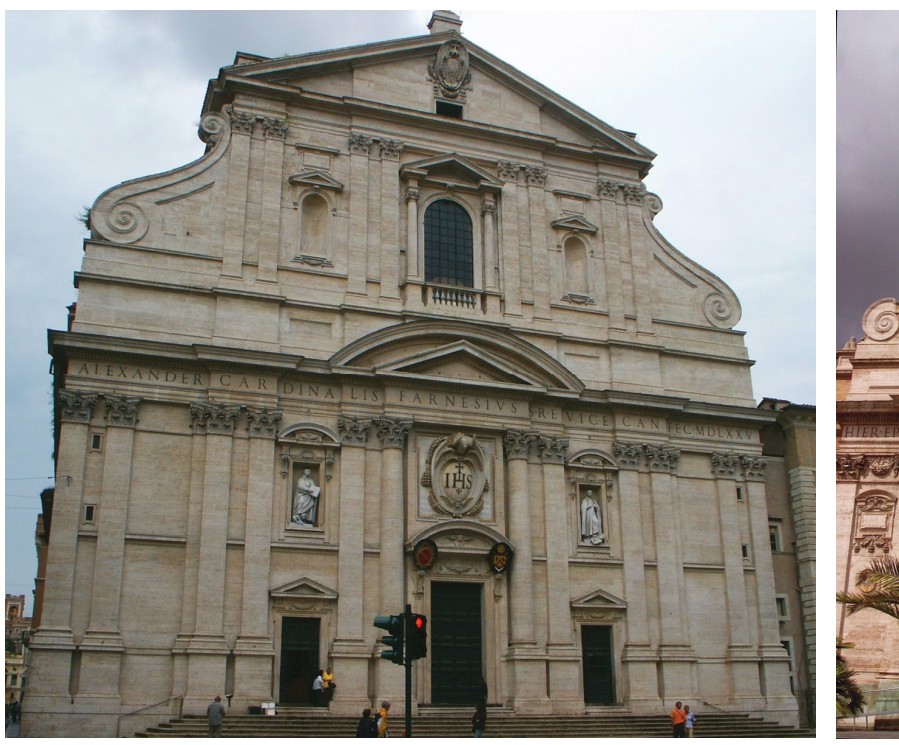 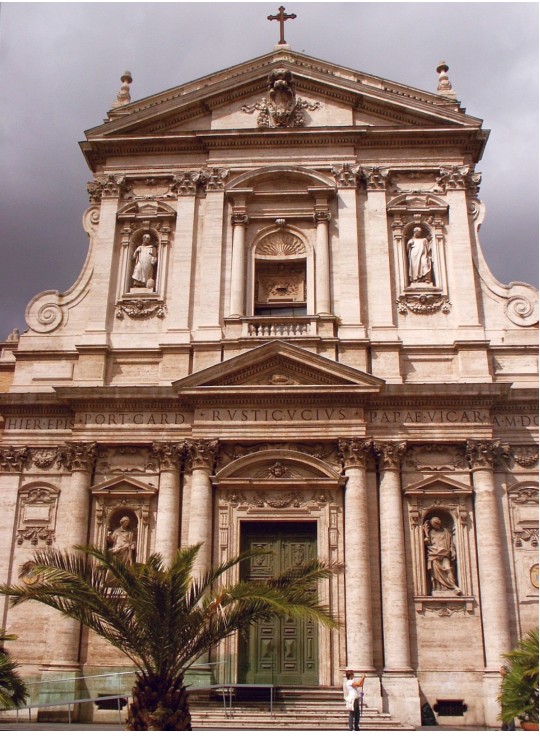

(**a**)                                                               (**b**)

**Figure 1.** Monochromatic façades made of travertine: (**a**) the façade of the Jesuit church Il Gesù (Giacomo Della Porta, 1584); (**b**) the façade of the church of S. Susanna in Rome (1597–1603). Photos from private collection.

## 4. Results: Solutions of Color Schemes of Façades in the Baroque Period

*4.1. Antique Heritage and Monochrome*

Ancient monuments, even if they once had polychrome, have been preserved in the colors of natural stone until modern times; surface damage has often obscured the original color of the material. Such a form was accepted as the canon in force in antiquity. Only the chiaroscuro gave diversified tonal shades. Printing techniques imposed a graphic black and white form of presentation of architectural patterns, first of all of orders in treatises from the earliest Renaissance (Serlio 1537; Palladio [1570] 1581), in illustrations to the works of Vitruvius and Alberti, and up to works from the end of modern times.

The architectural detail of monumental Baroque buildings in the early period was largely made of stone. In Rome, in most cases, a valuable material inherited from antiquity, travertine, was used (Bellini 2011), being carefully prepared to allow exposure of architectural forms. It can be seen in the realization of works starting from Mannerist period; particularly important for the development of sacral art, the façade of the Jesuit church Il Gesù (Giacomo Della Porta, 1584, Figure 1a), S. Susanna (Carlo Maderna, 1597–1603), or the most important building of the Catholic Church, St. Peter's Basilica (C. Maderna, 1603–1607, Figure 1b). Travertine became a distinctive feature of Roman buildings and was treated as an ideal material for shaping Baroque forms. Architects working in the Baroque period in Rome chose façades clad with this stone or, if funds were not sufficient, its imitation. In many cases, the whole surface of the façade was covered with stone. Such were the façades of churches designed by Giovanni Battista Soria, a pupil of Maderna, for the church of Santa Maria della Vittoria (1624–1626) (Acanto Restauri 2020a) and San Carlo ai Catinari (1635–1638), and the most famous architects of the period: Pietro da Cortona for SS. Luca e Martina (1635–1664, Figure 2a) or S. Maria della Pace (1656–1667); Francesco Borromini for San Carlo alle Quattro Fontane (1634–1667) and S. Agnese in Agone (1653–1657); Gian Lorenzo Bernini for San Andrea al Quirinale (1658–1661); and later, their collaborators and pupils, e.g., Carlo Rainaldi for San Andrea della Valle (1655–1663) (Acanto Restauri 2020b) or Santa Maria in Portico in Campitelli (1659–1667).

Some of those artists consequently opted for monochromatic solutions also in interior design. Pietro da Cortona always designed decorations of sacral interiors in light stone (mainly travertine) and stucco, e.g., the interior of the church SS. Luca e Martina (Castelli Gattinara 2015). Borromini also preferred such a coloristic conception (Blunt 1979), which is best seen in the case of the interior of San Carlo alle Quattro Fontane, where the rich chiaroscuro offers a diverse range of shades made of light stucco architectural details. When cheaper or more convenient solutions were applied, with only constructional parts of order systems made of stone, most often a lighter shade of plaster, almost identical to the color of stone, was chosen for the background of walls and sometimes covering of friezes, e.g., in the work of F. Borromini S. Ivo della Sapienza (Acanto Restauri 2020c) or the joint works of C. Rainaldi (1662–1677), C. Fontana, and G.L. Bernini (1678–1681), such as the twin churches in the Piazza del Popolo: Santa Maria dei Miracoli and Santa Maria in Montesanto (Benedetti 2012). In cases where the entire entablature was made of stone, especially when triglyph and metope friezes in the Doric order or inscription friezes were used, a two-colored rule of a different wall background with slightly contrasting detail was sometimes chosen (Figure 2b).

The influence of Roman architecture was strong throughout Catholic Europe during this period. However, local building traditions were simultaneously continued, and natural material resources were utilized. Other types of stone were used, both for exposed elements of architectural orders and for wall cladding, especially in designs influenced by Roman Mannerist and early Baroque patterns as well as those of the late Baroque. Marbles and limestones similar in color to travertine were chosen, e.g., in northern Italy, *pietra d'Istria* was commonly used in the cities of the region and Venice itself, e.g., on the façades of the temples of S. Maria della Salute (Figure 3a) and S. Moisè; in southern Italy, in Lecce, *pietra leccese*, was used on the façades of the churches of S. Croce, S. Angelo, S. Irene, Chiesa del Carmine; in Sicily, limestone from the area of Noto was used in the cathedral (Figure 4) and

the churches of S. Salvatore and S. Francesco in this city, or limestone from the vicinity of Syracuse in the cathedral and church of S. Spirito (Figure 3b).

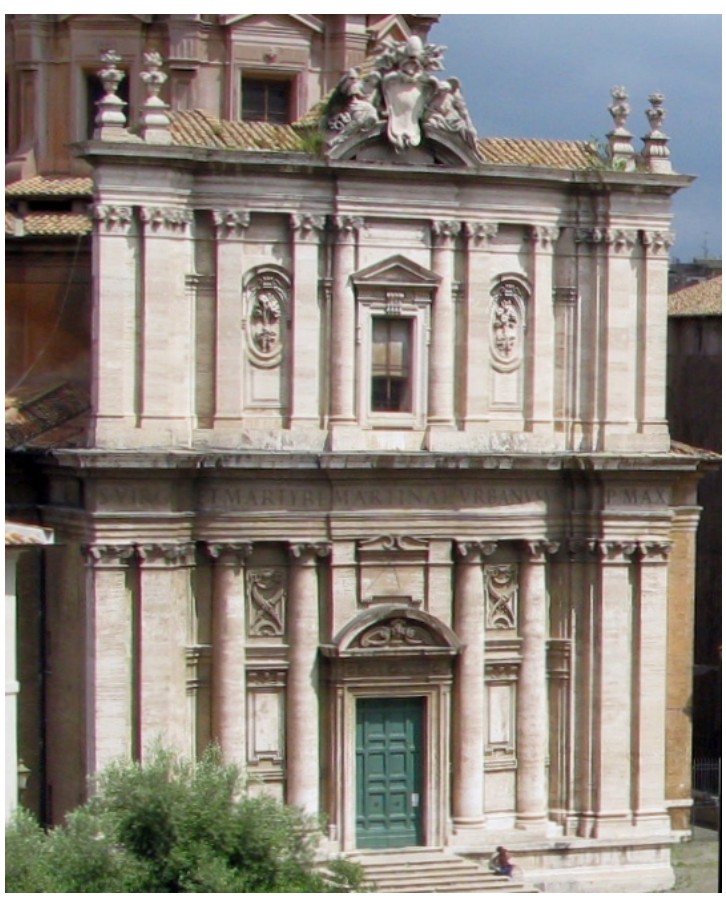 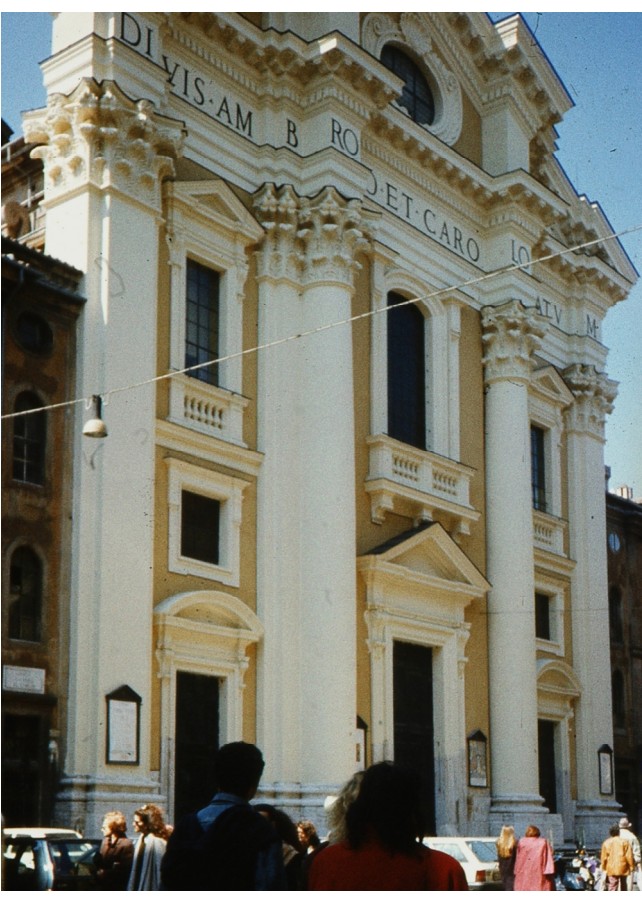

(**a**)                                                                                                   (**b**)

**Figure 2.** Monochrome architectural orders: (**a**) preserved small, balanced profiles within the details with subtle shadows, which ensured full visibility and clarity of the shapes. Cornices and architrave in the form of homogeneous stripes without ornamental detail. Fragment of the façade of the church of Santi Luca e Martina in Rome by Pietro da Cortona (1635–1664); (**b**) stone entablature with an inscription frieze and a combination of a different backgrounds on the façade of the church of the Lombard community in Rome, Santi Ambrogio e Carlo al Corso (1684), by Cardinal Luigi Alessandro Omodei. Photos by author and from private collection.

Similarly, outside Italy, for example, in central France, local limestone sandstone (*calcaire lutétien*), already used in Romanesque and Gothic buildings, was still applied to façades of churches of St-Gervais-et-St-Protais, Saint-Paul-Saint-Louis, Val de Grace, Les Invalides, or palaces in Versailles, Vaux-le-Vicomte, Compiègne, Chantilly, Écouen, Pierrefonds, and Coucy. In Lower Austria, Eggenburg sandstone (*Eggenburger Kalksandstein*) was used, for example, on the facade of the Dominican church in Vienna (Ecker 2010). In Innsbruck, a similar sandstone (*Hötting Breccia*) decorated the façade of the cathedral (Unterwurzacher and Obojes 2012, Figure 5a). In Lesser Poland (Małopolska), e.g., in Cracow, Dolomite and Jurassic limestone from the region was utilized on the façade of the Jesuit church of St. Peter and Paul (Olesiak 2017, Figure 5b). However, buildings were also faced with sandstones of different colors (against opinion, Koller 1998), such as the traditional dark stone *pietra forte* from Florence (e.g., the façade of the church of Santi Michele e Gaetano) or the multi-colored sandstone on the façade of the Jesuit church in Antwerp.

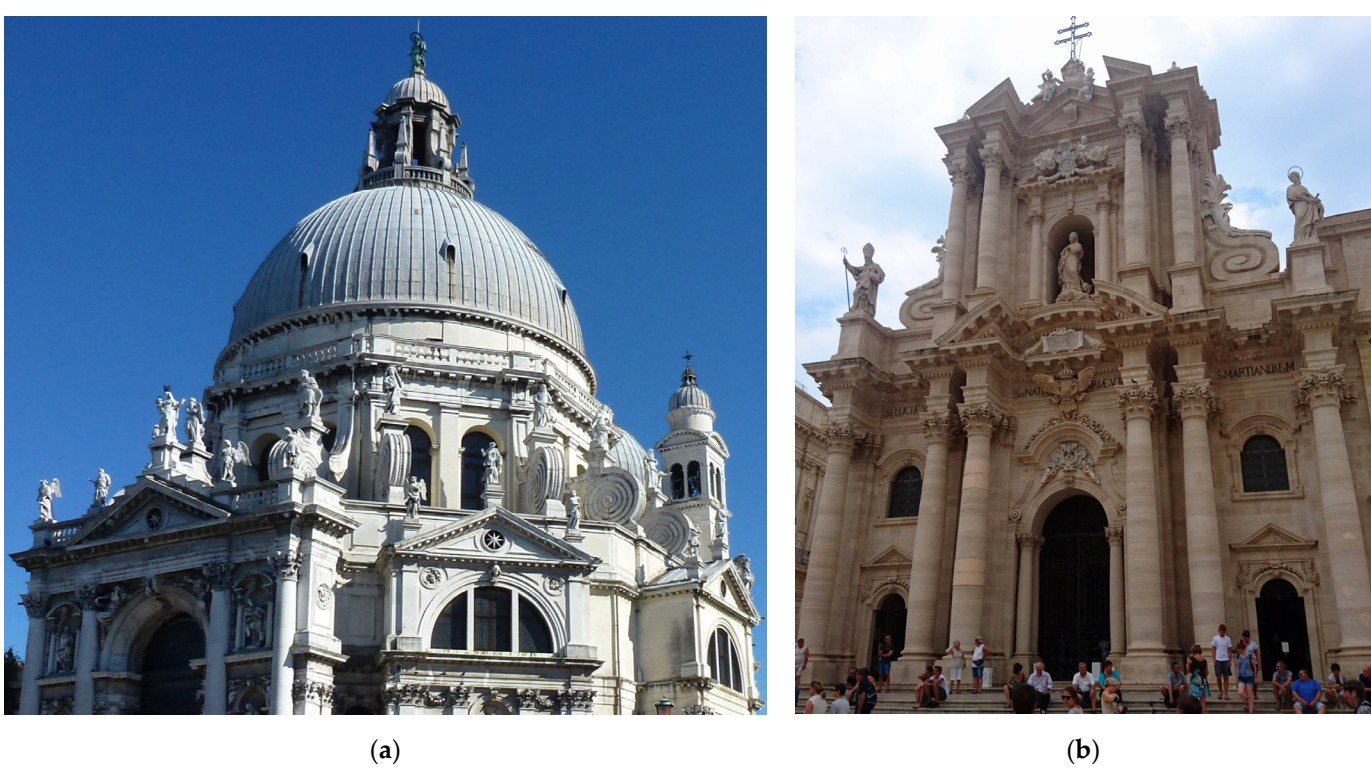

(**a**)　　　　　　　　　　　　　　　　　　(**b**)

**Figure 3.** Monochromatic façades (**a**) made of *pietra d'Istria*—the façade of the church of S. Maria della Salute in Venice; (**b**) made of local limestone—the Cathedral of the Nativity of the Blessed Virgin Mary in Syracuse (1725–1753). Photo by author and from private collection.

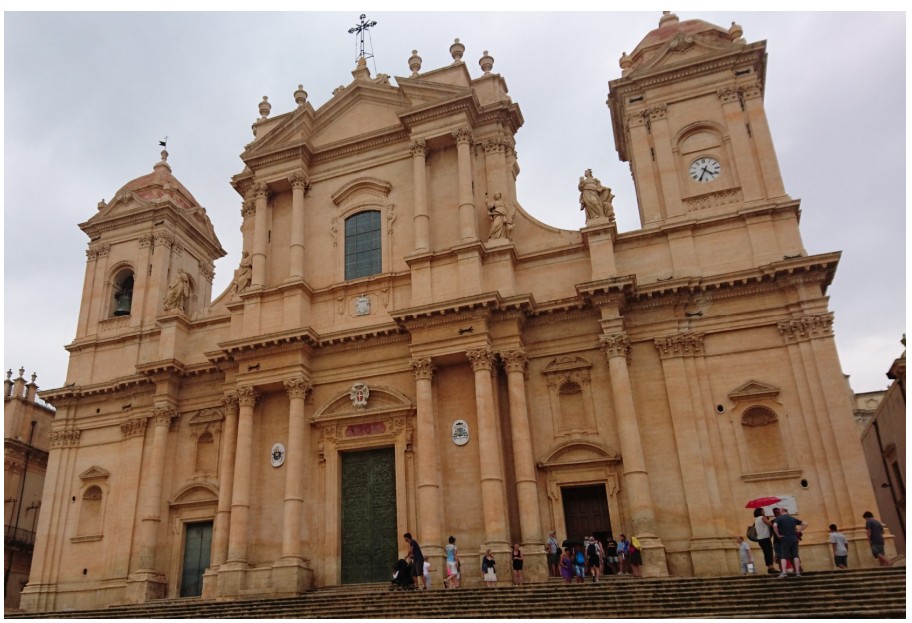

**Figure 4.** Monochromatic façade of the cathedral of Noto. Photo from private collection.

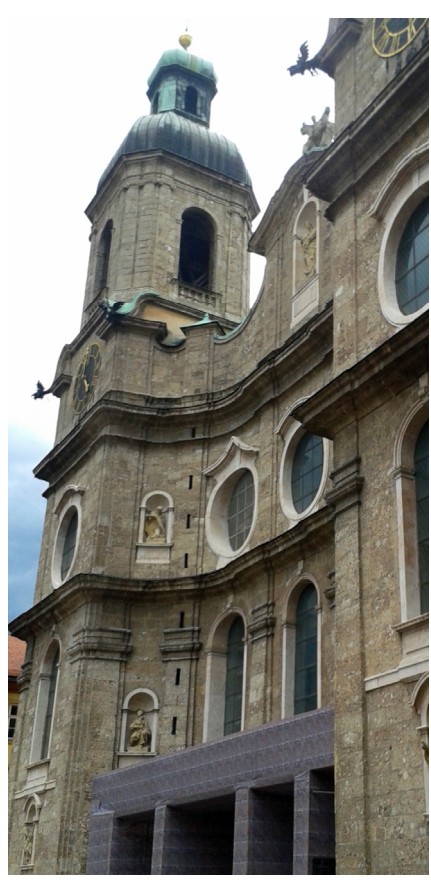 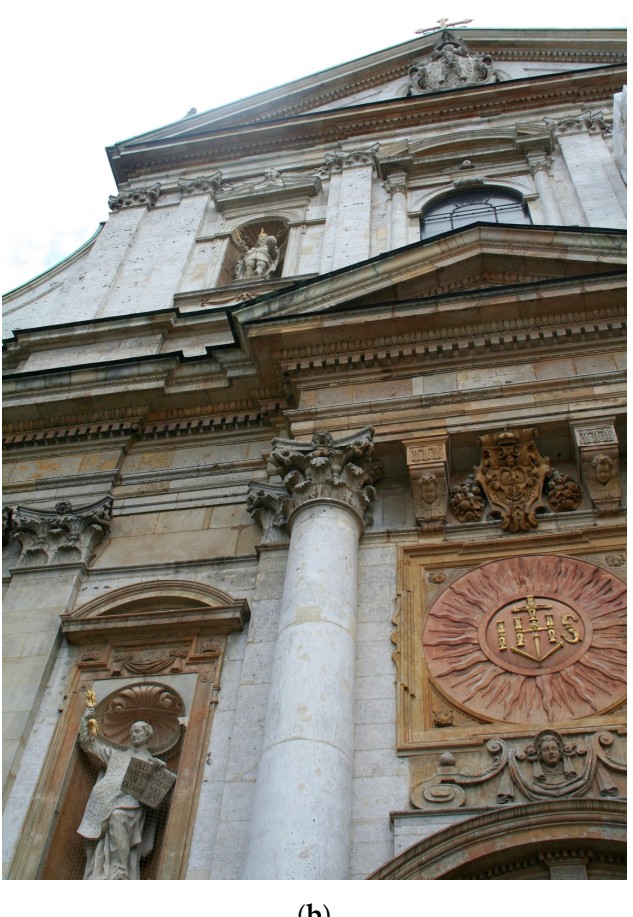

(**a**) (**b**)

**Figure 5.** Monochromatic or semi-monochromatic façade: (**a**) the façade of Innsbruck Cathedral of *Hötting Breccia*; (**b**) the façade of the church of Saints Peter and Paul in Cracow (1605–1619 to 1630) faced with Dolomite and Jurassic limestones from the region. Photos by author.

Monochromatic color solutions of architectural orders were also designed in the late, dynamic Baroque, drawing on the ideas of Borromini and Guarini. Single-colored stone cladding was used: light sandstone, as in the case of the façade of the Bamberg Jesuit church (Gunzelmann 2016) designed by Georg and Leonhard Dientzenhofer (1686–1693), the façade of the monastery church in Michelsberg by Leonard Dientzenhofer (1696), and the abbey church in Neresheim by J.B. Neumann (1747–1792). The darker iron sandstone (*Eisensandstein*) was used on the façade of the abbey church in Banz by Leonhard and Johann Dientzenhofer (1698–1719) and the pilgrimage basilica of the Fourteen Holy Helpers near Bad Staffelstein, built according to plans by J.B. Neumann (1743–1772), or even red sandstone was employed, as on the façade of the Jesuit church in Heidelberg (1712–1759) and the façade of the Neumünster abbey church in Würzburg (1711–1716), the design of which is attributed to Johann Dientzenhofer. An exceptional solution for a façade is the uniform brick face of Palazzo Carignano in Turin by Guarino Guarini (1679), who consciously used this material in a manner customarily reserved for precious stones (Piccoli 2012). His imitations only appeared in the late Baroque designs of Bernardo Antonio Vittone.

Façades were also covered with almost monochromatic colored elements made of various stone and stucco materials. Thus, the façade of Salzburg Cathedral (1614–1628) by Santino Solari was covered with so-called Untersberg marble using marbling and additions of light limestone. Sometimes a decision was made to unify the color of the natural light stone face entirely with a lime coating, as in the case of the façade of Pasava Cathedral (1668–1693, Figure 6) by Carlo Lurago (Hauck 2011). The court architects from Vienna used this

façade finish particularly often, e.g., for the façades, designed by Lukas von Hildebrandt, of the Maria Trau Piarist Church in Vienna (1698–1719), the so-called Reichskanzleitrakt at the Hofburg residence (1722–1726) (Koller 1997, 2003a), the Stadtpalais Prinzen Eugen (Winter Palace of Prince Eugene, 1697–1724, Johann Bernhard Fischer von Erlach and Hildebrandt), or the court church of St. Charles Borromeo (after 1716, J. B. Fischer von Erlach). Sometimes darker ochre tones were chosen, as in the case of the exterior façades of the Esterház Palace (1663–1672, designed by F. Lucchese and executed by artists of the Carlone family). Christoph and Kilian Ignaz Dientzenhofer chose a lighter shade, as in the case of the façades of the Abbots' Church in Břevnov (1708–1740) or in Legnickie Pole (1719–1729, restored in the last renovation between 2014 and 2017), the churches of St. Nicholas in Mala Strana (Figure 7), St. Nicholas in the Old Town, or St. John of Nepomuk on the Rock in Prague, performed in collaboration with the stonemason František Santini-Aichel (Kotrba 1976). In some cases, slight tonal differences remained between stone elements and stucco or plaster.

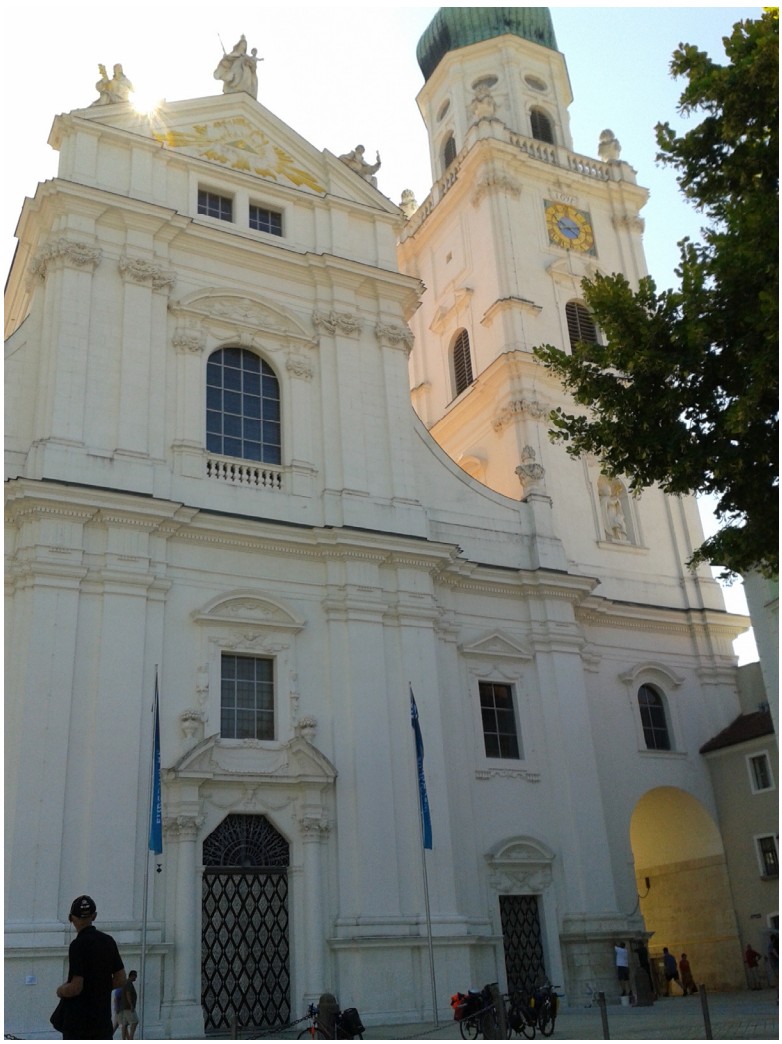

**Figure 6.** Unification of the color of the natural light stone face covered entirely with a lime coating: the façade of Pasava Cathedral (1668–1693). Photo by author.

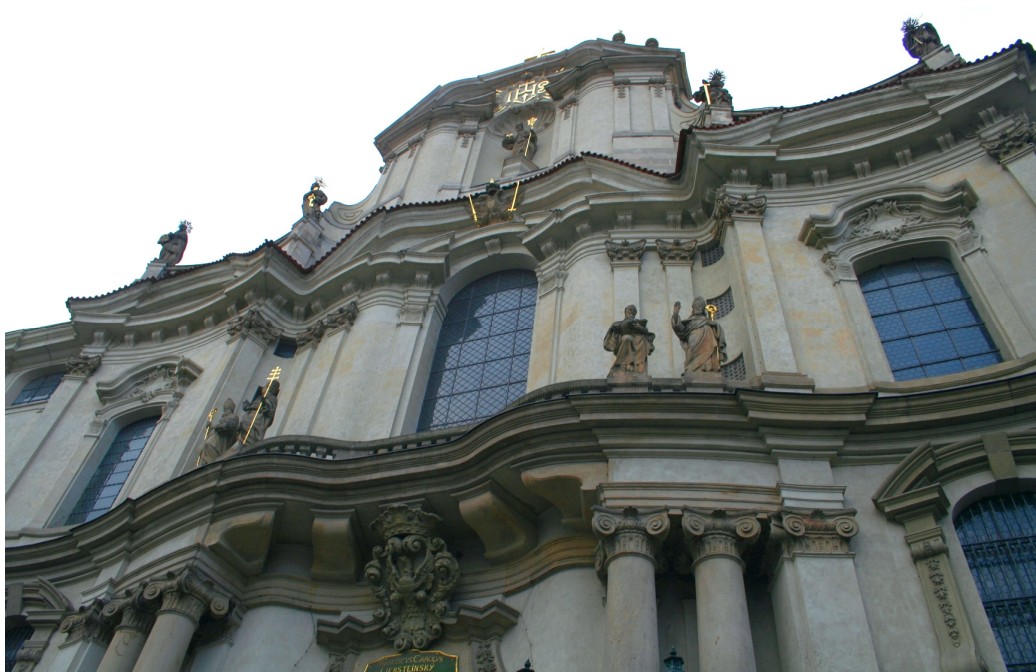

**Figure 7.** Strongly embattled entablature in the works of Christoph and Kilian Ignaz Dientzenhofer. Facade maintained in almost monochromatic colors with elements of various materials: stone and stucco. Facades of the churches of St. Nicholas on Mala Strana of Prague. Photo author.

Often, however, when stone construction elements were complemented with stucco ornaments or when only brick construction was used and covered with plasters and stuccowork, bichromatic solutions were introduced in colors. Usually, architectural details made of stone or stucco were unified in one color that stood out against a plaster or brick background, sometimes only with chapiters of pilasters made in another material. Plasters imitating the colors of stone or brick were used. Some solutions are associated with early Renaissance Florentine concepts, such as the façades of Jesuit churches in Innsbruck (1627–1646, Figure 8a), where dark sandstone elements were exposed against a white background, or in Lucerne, where additionally granite capitals were used (Storemyr 2001). More subtle color combinations dominated; light, white, cream, or ochre architectural details, made of stone or its stucco imitation, appeared on the light background of plasterwork. Sometimes a white background was chosen, e.g., on the façade of the Jesuit church in Vienna (1624–1631) designed by Giovanni Battista Carlone and the façades of Gartenpalais Liechtenstein in Vienna by Domenico Egidio Rossi and Domenico Martinelli (1688–1700). More often a pastel-colored plaster was used: grey, as on the façade of the former convent church of Garsten (1685) designed and executed by Pietro Francesco Carlone and his sons, and even pink, as on the façades of the Jesuit church in Pasava (1677, P.F. Carlone, Figure 8b) (renovation 2008) or green, on the façades of the Jesuit church in Linz (1669–1678, P.F. and C. A. Carlone). Probably the most frequently used color of plaster was ochre yellow, as in the case of the walls of the University Church in Wrocław (1689–1698). At the beginning of the 18th century, this two-colored scheme of polychrome of external façades with the use of yellow ochre or brick-red shades (Koller 2003b, 2007) began to dominate in southern Germany, which was influenced by the choice of color decoration for the Imperial Schönbrunn Palace designed by Fischer von Erlach (Koller 2003c; Rohatsch 2005) (see Appendix A, A3).

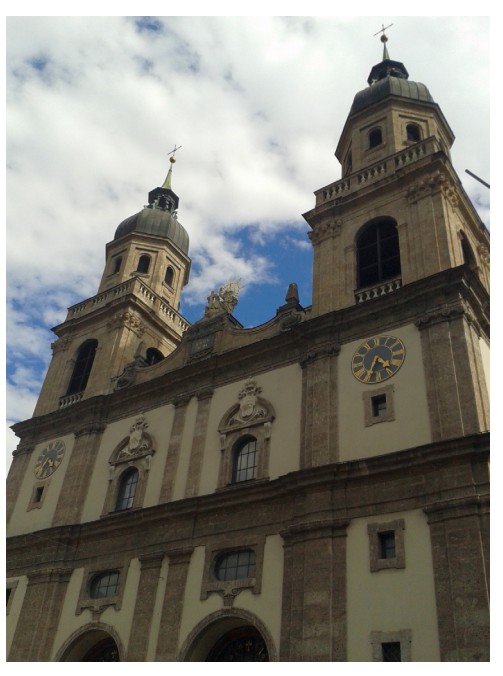
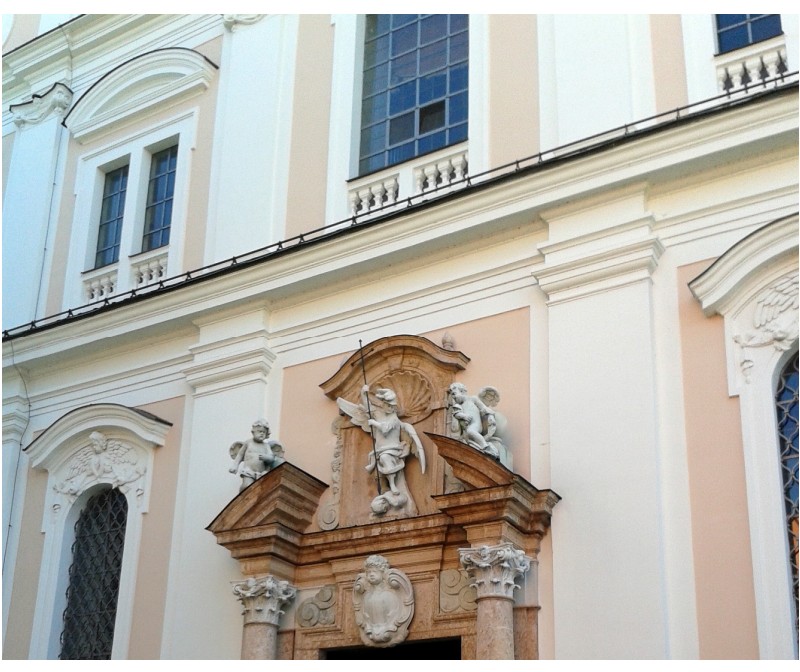

(**a**)　　　　　　　　　　　　　　　　　　　　　　　　　　　　　(**b**)

**Figure 8.** Monochromatic architectural order on colored walls: (**a**) creped entablature in sandstone architectural order—the façade of the Jesuit church in Innsbruck (1627–1646); (**b**) white order on pink walls—the façades of the Jesuit church in Pasava (1677). Photos by author.

*4.2. Color Breakdown of the Horizontal Entablature: Distinguishing the Colour of the Frieze*

In classically elaborated order decorations, an inscription frieze, often with gilded or dyed letters, sometimes replaced with an ornamental belt, became an artistic element diversifying architecture. Such friezes can be observed both on façades and in interiors of the most famous Roman churches, such as St. Peter's Basilica and the churches of Il Gesù, S. Andrea della Vale, or S. Ignazio. The frieze, like other elements of the entablature, was made of stone, with engraved or carved surfaces. Solutions using brick and stucco replaced expensive materials, usually imitating them with colors. Architectural orders treated in this manner became popular, especially with the so-called Jesuit façade pattern. On most of the church façades designed according to this model, inscription friezes were used in the main entablature, similar to triumphal arches.

When various materials were used for exterior wall structures and their plaster, stucco, and paint finishes, and particularly when materials for the entablature were varied, it was common to treat the frieze, and sometimes also the shafts of the supports, as a background, contrasted with cornices, architraves, and chapiters. Both in the Mannerism and early Baroque, slight chromatic differences were used. In one of his first architectural works, the façade of the church of S. Bibiana in Rome (1624–1626), Bernini used such an arrangement with a light sand color of plaster in the frieze area between the travertine cornice and architrave and the same on the shafts of pilasters (Ticconi 2020). Borromini designed the same color arrangement for the façade of the Oratorio di San Filippo Neri in Rome (1637–1667), using cheaper techniques, due to the requirements of the monastic rule (Argan 1955; Portoghesi 1967). This was also how Francesco Maria Richini (Ricchino) arranged the colors of the façades of churches in Milan after his stay in Rome (churches of San Giuseppe, Santa Maria alla Porta, 1652). The architect-decorator Carlo Lurago, a native of the Como region who moved to Bohemia, used rich stucco detail in friezes and on chapiters, as on the façades of the Prague churches of the Most Holy Saviour in Clementinum (1638–1648) and St. Ignatius in the New Town in Prague (1665–1670, Figure 9a). In Tirol, entire façades were often polychromed in white, and the architectural order was decorated in yellow with a white frieze, an example being the pilgrimage church near Salzburg Maria Plain (Giovanni

Antonio Dario, 1674). On the façade of the church of St. Teresa in Vilnius (1654, Figure 10a), attributed to Constantine Tencalla from the Ticino region, the architectural order with stone capitals is distinguished by white belts of cornices and entablature and a dark frieze; St. Anne's Collegiate Church in Cracow (Figure 9b) has cream-colored walls with sandstone and limestone detail (Tylman of Gameren 1689–1695).

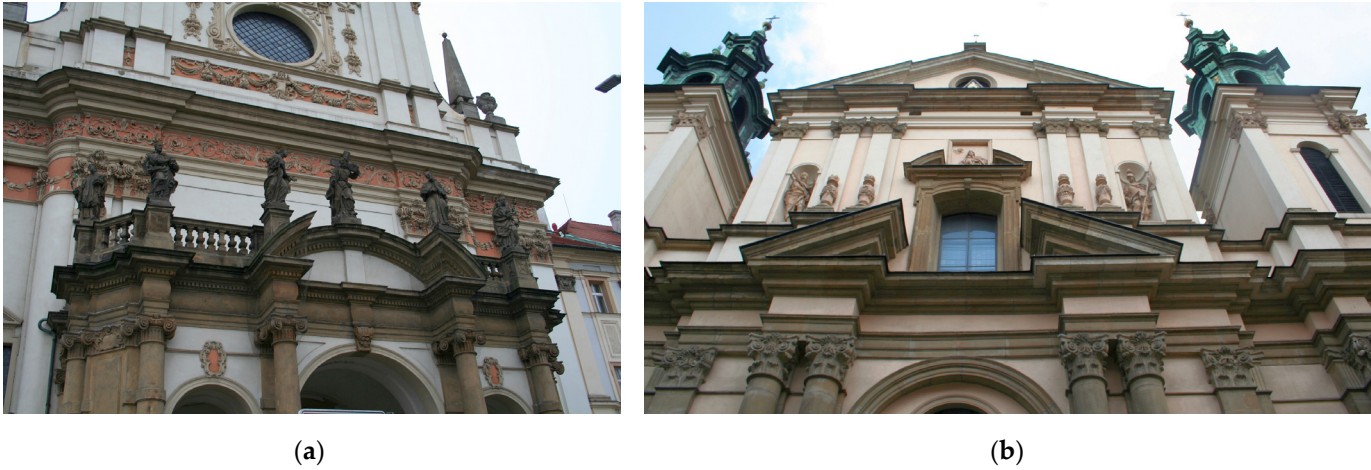

(**a**)                                                    (**b**)

**Figure 9.** Entablature with a frieze of another type of stone or stucco: (**a**) colorful ornamental frieze in Carlo Lurago's architecture. Fragment of the façade of the church of St. Ignatius in the New Town in Prague (1665–1670); (**b**) St. Anne's Collegiate Church in Cracow, with cream-colored walls with sandstone and limestone detail. Photos by author.

In the eighteenth-century Baroque, as mentioned, architects, especially Viennese court architects, most often used uniformly colored entablature, and more often whole façades in white or cream. However, J.B. Fischer von Erlach, when designing the façades of the churches in Salzburg, used a subtle two-tone arrangement with a frieze in the color of the walls; in the case of the façade of the Holy Trinity Church (1694–1702), a white order on a light grey background; and the Kollegienkirche (Collegiate Church) (1696–1707, Figure 10b), a grey order, light background, and stucco decorations (Koller 1998). Kilián Ignác Dientzenhofer proposed a similar solution for the façades of the Church of St. John of Nepomuk in Prague-on-the-Hradany (1720–1729) and St. Magdalene in Karlovy Vary (1732–1736). The façade of the Priesterseminarkirche in Linz, designed by J.L. von Hildebrandt (1718) and completed by the builder Johann Michael Prunner in 1725, was painted in equally subtle but warm tones with a white background and friezes. The monastery church in Jablonné (Gabel, 1699, Figures 11 and 12), designed by Hildebrandt, was instead given a colorful façade (perhaps a contribution by Domenico Perini, who completed the construction). In the Asam brothers' designs, for example, the façade of their church in Munich, the so-called Asamkirche (1733–1746, Figure 12a), was decorated with marbled friezes, with orange color between the white elements (Reichwald 1976). Another façade of the Maria de Victoria Church in Ingolstadt, built according to their concept, was similarly colored. Such two-colored combinations within the entablature and usually different colored shafts of supports were used by late Baroque Italian architects, e.g., in Rome: Giuseppe Sardi for the Church of S. Maddalena (1735); in Naples: Luigi Vanvitelli, Ss. Annunziata (after 1757), and Ferdinando Fuga for the church of dei Girolamini (1780), Basilica di Santa Maria degli Angeli a Pizzofalcone (2nd half of the 18th century); in Turin: Carlo Juvarra for the Church of Santa Cristina (1715–1718). Juxtapositions with sand background and white elements typical of Southern Germany and Lower Austria were used, as on the façades of the monastery in Klosterneuburg (confirmed according to the renovation of 2013) (Ecker 2013), and were repeated the other way round, with colorful architectural details and with sand yellow on a white background, as on the façades of the palace in Wilanów (Jaskanis 2011), orange-pink on Basilika Maria Dreieichen (1733)

near Mold, or the parish church in Schwechat (1765) in Lower Austria (Koller 2007). There were also examples of façades using shades of green, as on the façade of the Fürstenfeld monastery church (1701–1747, Giovanni Antonio Viscardi, Johann Georg Ettenhofers, Figure 12b).

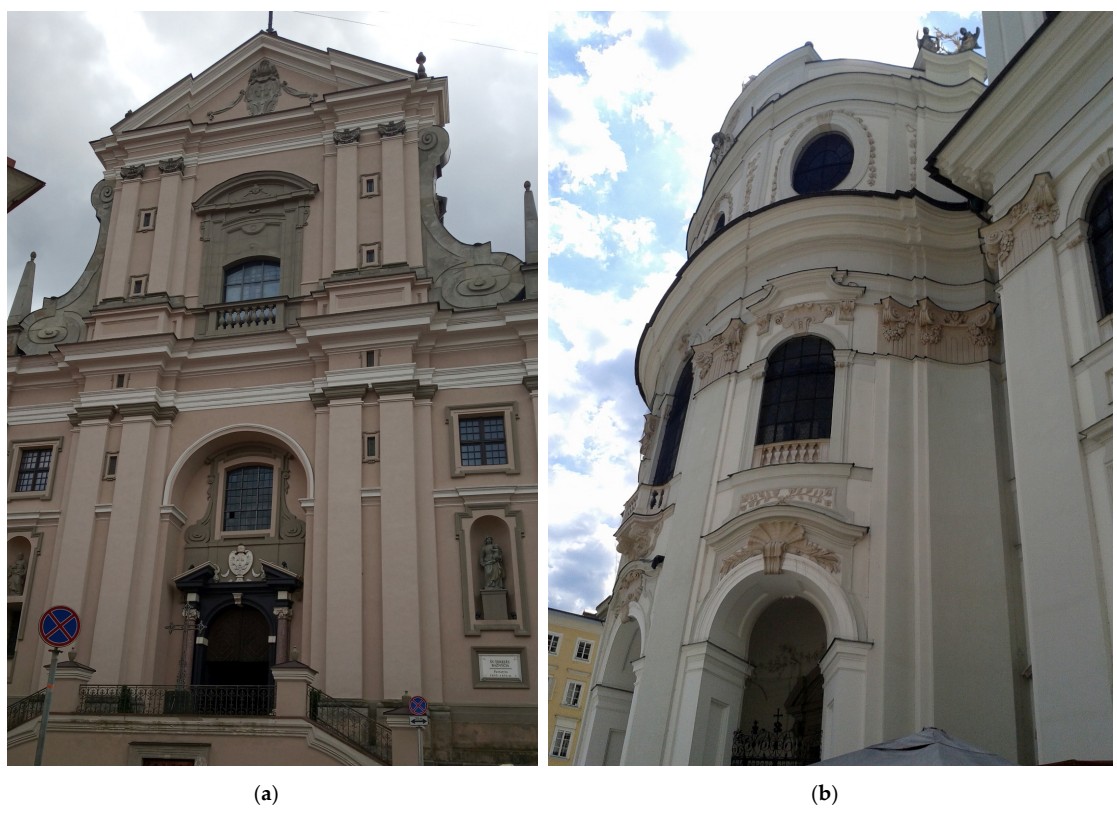

(**a**)  (**b**)

**Figure 10.** The architectural order with stone chapiters is distinguished by white stripes of cornices and architrave and a darker frieze: (**a**) the façade of the church of St. Teresa in Vilnius (1654); (**b**) a grey order, light background and stucco decorations—the Kollegienkirche (Collegiate Church) by J.B. Fischer von Erlach (1696–1707). Photos by author.

In some circumstances, architraves were also a different color than the rest of the entablature elements, which usually resulted from differences in the materials. Initially, this was the case with multi-colored elements of the entablature in the continuation of Renaissance and Mannerist solutions, e.g., in Milan, where differently colored stone elements of red, yellow, and white were used. A few examples can also be found in the late Baroque. Probably due to the material used, especially spolia, a distinctive colorful architrave was introduced on façades at that time. Examples from Catania can be invoked here: a granite architrave in the Cathedral of S. Agata (Figure 13) or local light stone, the so-called *pietra giurgiulena* (*pietra bianca di Siracusa*), in the church of Santa Maria dell'Elemosina.

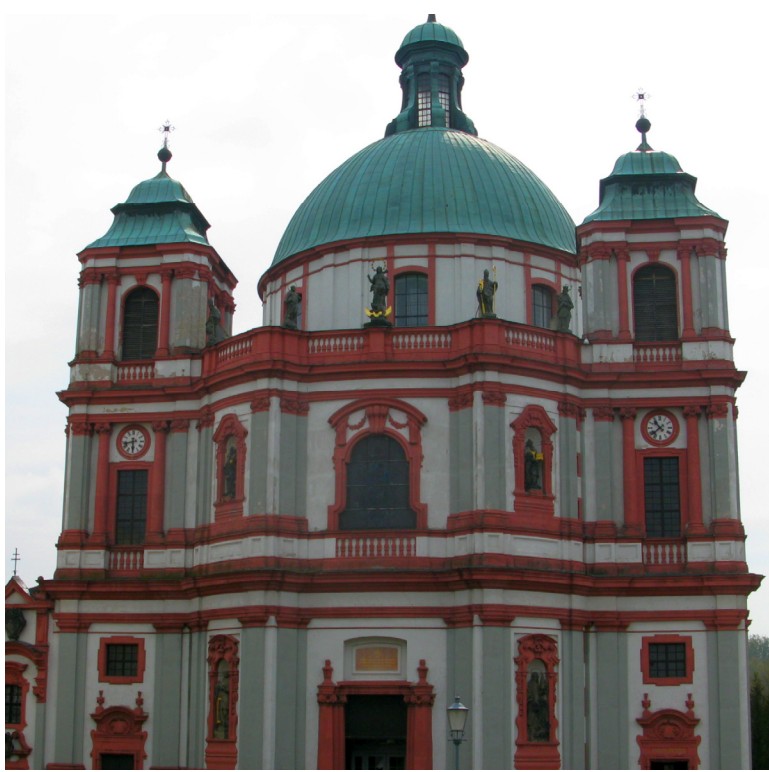

**Figure 11.** The colorful façade of the monastery church in Jablonné (Gabel, 1699), designed by J.L. von Hildebrandt. Photo from private collection.

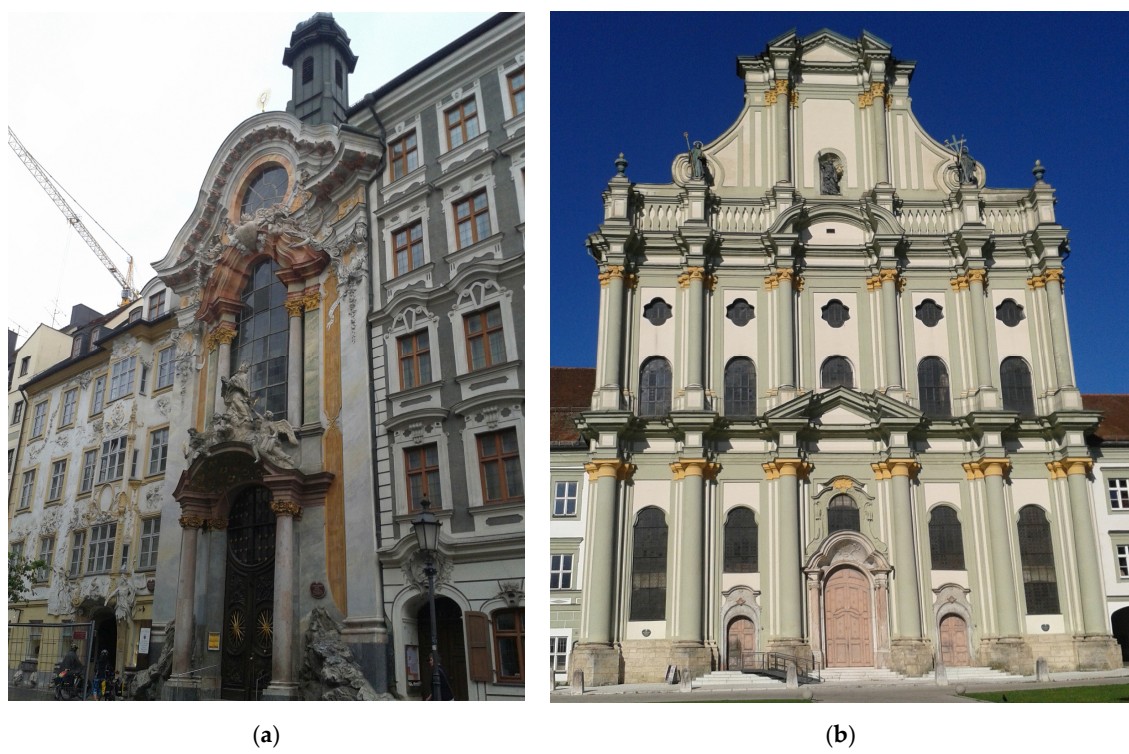

(**a**)  (**b**)

**Figure 12.** Two-color entablatures: (**a**) made in the marmorisation technique—the façade of the church of St. John of Nepomuceno in Munich, the so-called Asamkirche (1733–1746); (**b**) colored in shades of green—the façade of the monastery church Fürstenfeld (1701–1747). Photos by author.

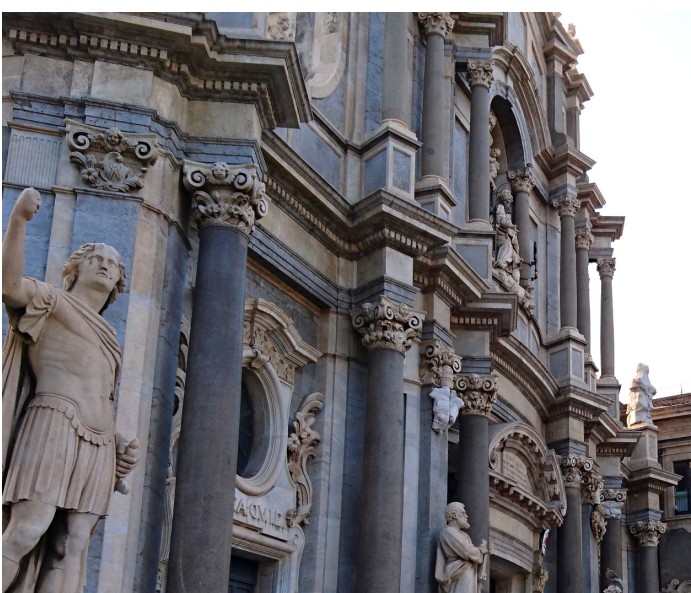

**Figure 13.** Distinctive colorful architrave introduced on façades and granite architrave in the Cathedral of S. Agata in Catania. Photo from private collection.

*4.3. Color Breakdown of the Frieze in Vertical Arrangement: Colors of Pseudo-Imposts*

The embattling of the cornice changed the way the frieze was made; it was necessary to strengthen a fragment of the frieze above the capitals, and thus the frieze was made entirely of stone or strengthened in the part where the frieze was fragmented. Initially, efforts were made to obliterate the traces of differentiation. However, with time, the introduction of stone elements within the frieze that differed from the rest of the plastered frieze resulted in a change of color of these fragments. Such construction techniques appeared in northern Italian regions, including Lombardy. This may have resulted from the implementation of local construction traditions in classical architectural solutions, but mostly was due to attempts to save precious building materials, which in this region was stone. F.M. Richini, the author of various versions of the Milanese Cathedral façade designs based on Roman models, introduced this type of structure into the design of the façade of the church of San Giuseppe (erected between 1629 and 1630) and later into the façade of the church of S. Martino in Veduggio con Colzano (1642, visible after conservation in 2012) (Balestreri 2017; Wittkower et al. [1958] 1999, p. 86). His collaborator and successor on the Milan Cathedral site, Carlo Buzzi, followed a similar approach in numerous façade designs for churches in Milan and the surrounding area. At the same time, Jesuit architects who based their projects on the Tibaldi model applied such constructional and polychrome concepts. This is best seen in the case of the retention of unplastered stone elements, as in the case of the façade of the Jesuit church in Lviv (ca. 1630) by Giacomo Briano, a Jesuit from Modena, where the structural elements, including the reinforcement of the frieze over the chapiters of the Corinthian pilasters, were made of dark stone, contrasting with the white of the plastered walls (Paszenda 1972, 1973, 1999; Wittkower et al. [1958] 1999, p. 97; Betlej 2002) (A4). The architects and builders from Como and Ticino regions had similar designs. Here we can mention the work of stucco artists and builders of the Lurago family. On the stucco façades of the churches of San Giorgio (c. 1647–c. 1685) and San Barnaba (after 1660) (Iton S.r.l. (Interventi speciali per l'edilizia) 2018; Angelo 2018) in Modena, Antonio Loraghi (Lurago) (Pinto 2006; Bieri 2017) designed a composition with continuous vertical elements, contrasting white on brick backgrounds. His older brother, Carlo Lurago, active in Lower Austria and Bohemia, worked in the same way on church and monastery façades, but chose lighter, less contrasting backgrounds, as in the case of the pilgrimage church façade in Maria Taferl (1670–1671), the church of the Assumption of the Virgin Mary in Hradec Králové (1654–1679, Figure 14a) or the façade facing the river in Prague's Clementinum (1654–1679).

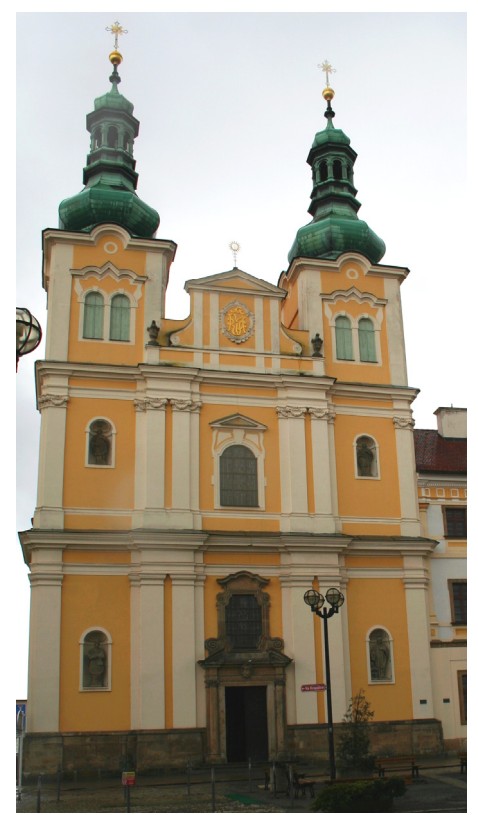
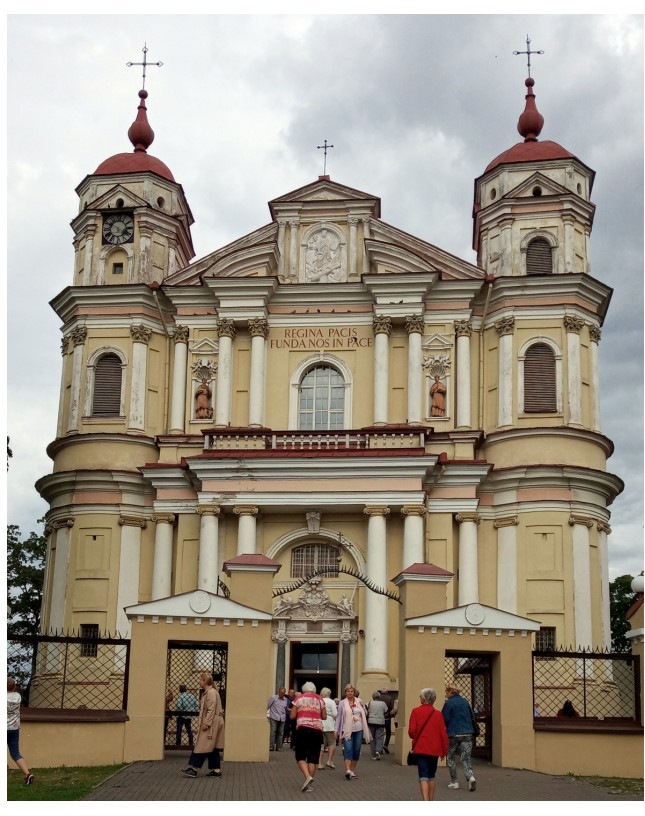

(**a**)　　　　　　　　　　　　　　　　　　(**b**)

**Figure 14.** Composition with continuous vertical elements in architectural orders: (**a**) the façade of the church of the Assumption of the Virgin Mary in Hradec Králové (1654–1679) by C. Lurago; (**b**) the façade of the church of St. Peter and Paul on Antokol in Vilnius (1670s). Photos by author.

Simultaneously with the development of the dynamics of architectural forms in the Baroque, compositions were made more vertical. This was connected with interest in Gothic heritage, present in the works of Northern Italian Mannerists and their continuators, e.g., Borromini himself, taken up by Baroque artists working in territories with strong Medieval traditions. This manifested in the use of architectural profiles referring to Gothic forms, sometimes with literal quotations (best seen in the works of, e.g., J.B. Santini Aichel) but also in the use of structural solutions (e.g., Czech Dientzenhofers implemented a solution with buttresses inside). However, the most visible manifestation was accentuation of façades with strongly visible, often multiplied elements of vertical divisions: half-columns and pilasters or lesenes. Such solutions were of particular importance in the works of mature and late Baroque architects working in Bavaria, the Habsburg Empire, Bohemia, and Silesia, and later their students and followers, who came to the Grand Duchy of Lithuania.

Local builders who collaborated with northern Italian artists continued these ideas. This was the case with the façade of the Jesuit Church of St. Francis Xavier in Lucerne (1673–1677), which may have been designed by Tommaso Comacio of Roveredo in Grisons (Santi 2004), but was built by Michael Beer and Michael Thumb from the province of Voralberg and completed by Jesuit Heinrich Mayer (Bieri [2008] 2012). The same artists reproduced this design on the façade of the Premonstratensian church in Obermarchtal (1686). H. Mayer also designed the façade of the church of his order in Soleure (Solothurn) in Switzerland (1680–1687) (Carlen 1981). Sometimes, a version emphasizing even more the vertical arrangement of the façade's structure was chosen by resigning from the continuous architrave and frieze elements and replacing them with the impost, supporting the cornice above the pilasters' chapiters, as on the façade of the Am Hof church in Vienna (1662) by C.A. Carlone or some works by Czech Dientzenhofers.

The continuity of color in the elements of the vertical structure, i.e., the painting of part of the frieze, discontinued above the pilasters, in the same color as the supports themselves, has been frequently reconstructed recently during conservation works and restoration of the color scheme of church façades. Such solutions were most often chosen by architects or builders from northern Italian and Swiss regions who cooperated with them. One can cite examples from the 1680s to 1690s erected throughout Central Europe from the region of Upper Austria, including façades in Steyr: of the Dominican Marienkirche (1642–1644), the Jesuit St. Michael's Church (1648–1677), and the pilgrimage church of Heiligenkreuz near Kremsmünster (C.A. Carlone, 1687–1690); Lower Austria, e.g., from Vienna: the façade of the pilgrimage church Mariahilf under the care of the Barnabites (until 1711) (with coloring reconstructed in 2003) (Ecker 2003b); Silesia, e.g., from Wrocław: façades of the churches of St. Jacob (now St. Anne, Sigmund Linder, 1686–1690) (maintenance works 2010) and St. Anthony (1685–1692) (restoration 2011); Poland: the façade of the church of the Oratorians near Gostyń (Jerzy and Jan Catenazzi, 1675–1698), the church erected for the Jesuits in Krasnystaw of St. Francis Xavier, (Jan Ignacy Delamars, 1697–1715) (Rusińska-Kurzątkowska 1956; Szewczyk and Bubicz 2020), the pilgrimage church in Święta Lipka (1687–1695, Figure 15a) built (until 1730) or perhaps designed by Jerzy Ertlie (Paszenda [1998] 2015) (restored to its original colors in 2010–2013) (Dzieciątkowska 2010); Lithuania, e.g., the façade of the Church of Saints Peter and Paul in Antokol in Vilnius (1670s–1770s, Figure 14b), designed by the Cracow architect Jan Zaor, probably modified by Giovanni Battista Frediani, who may have also designed a similar façade for the Trinitarian Church in that city (1694–1716) (Guttmejer 2019).

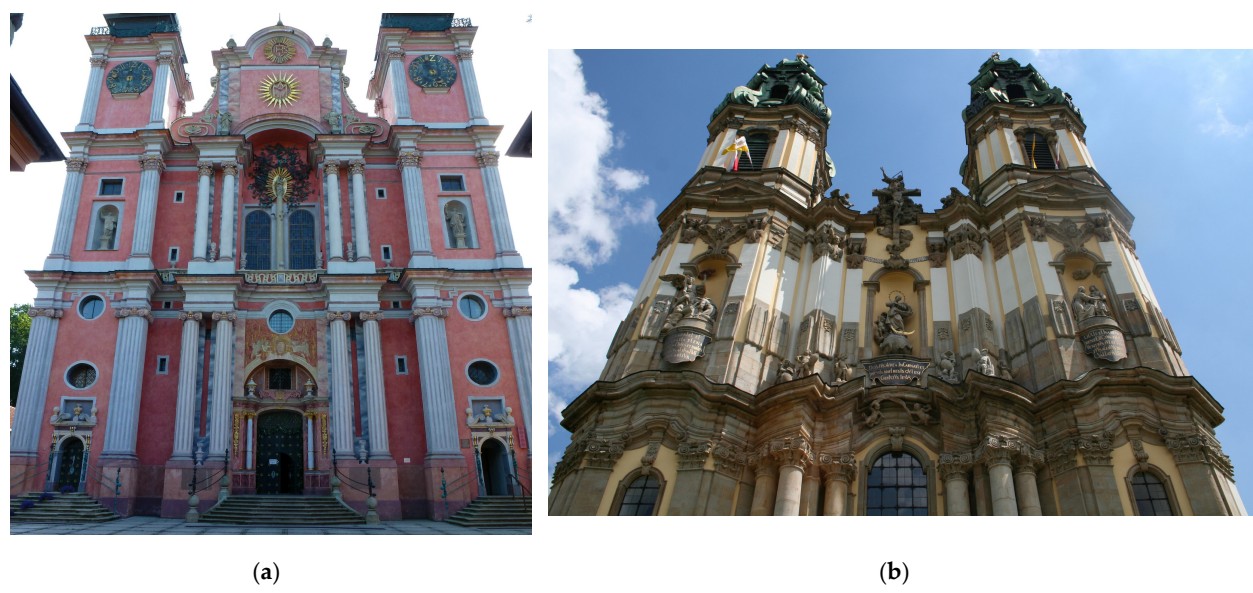

(**a**)                                                                                          (**b**)

**Figure 15.** Continuity of multiplied elements of vertical division emphasized by color: (**a**) the façade of the church in Święta Lipka (1687–1695); (**b**) the façade of the monastery church in Krzeszów in Lower Silesia (1728–1733). Photos by author.

The expression of the work shaped in this way could be highlighted by the discussed color scheme of the architectural order. The earliest examples of polychrome decoration of an entablature with a distinction of pseudo-imposts, different in color from the frieze, appear among the designs by Christoph and Kilian Ignaz Dienzenhofer on the façade of the monastery church of St. Joseph in Obořiště (1702–1711) and by Matthias Steinsell (Steindl) on the façade of the parish church in Laxenburg near Vienna (1703) (Ecker 2003a). They use slightly contrasting juxtapositions with light detail and sandy ochre colors. The monumental façade of the Loretto ensemble in Prague (Figure 16a), overshadowing the Church of the Nativity of Our Lord with the Loretto House (1722–1726) by the Dienzenhofer family, was performed in a similar layout using interrupted friezes and architraves

(imposts) (Líčeníková 2017; Koberová 2017). However, in many works of artists or builders from this circle, such a solution of polychrome is reproduced on continuous, only embattled entablature, as in the case of examples from Lower Silesia: the façade of the monastery church in Krzeszów (1728–1733, Figure 15b) (Ałykow 2014), Lubomierz (1728–1730, builder J.J. Scheerhofer, Figure 16b), the parish church of St. Valentine in Lubiąż, and the church of the Blessed Virgin Mary at the Cistercian monastery in Neuzelle in Lower Lusatia (c. 1730). Such polychrome arrangements of architectural order appear in the works of Jakob Prandtauer (1660–1726) on the façades of the abbey church in Melk (1711–1725) (Koller and Paschinger 1980; Koller 2010; Murczek 2017), the Carmelite church in St. Pölten, the pilgrimage church in Sonntagberg (1718–1732), and the parish church in Wullersdorf (1725–1733). They can be found among the works of Hildebrandt and artists from his circle, as on the façades of St. Peter's Church in Vienna (1701–1733), possibly in its final form designed by Kilian Ignaz Dientzenhofer, the Trinitarian Church in Bratislava (1717–1727), designed by Franz Jangel and J.L. von Hildebrandt, St. Anne's Church in Pilsen, Bohemia (1717–1727) by Jakub Auguston, who probably also used similar designs for the façades of monasteries, e.g., in Chotěšov, and palaces, the monastery church in Fürstenfeld by Giovanni Antonio Viscardi and Johann Georg Ettenhofer, the monastery church of St. Lawrence and St. Stephen in Innsbruck (1713–1719) by Georg Anton Gumpp (1682–1754), the church of the Brothers Hospitallers in Linz (1729–1732), and the Lamberg palace in Passau (1724) by Johann Michael Prunner. This color scheme was also used by the Graz court architect Andreas Stengg and his son Johann Georg Stengg, for example, on the façades of the Mariatrost pilgrimage church (1714–1724) and the Church of the Brothers Hospitallers in Graz (until 1740). In addition to these two-color systems with light architectural details and wall backgrounds in ochre, sometimes beige, or the other way around, with darker details on a light background, we can also find three-color polychromes with the use of colors unprecedented in the earlier period, such as in the Lower Austrian pilgrimage church in Mariahilfberg near Gutenstein (1724–1727), where a pink frieze between the white cornice and architrave and grey pilaster shafts were used (Koller 2007).

Such a color scheme of architectural order structure was commonly used by late Baroque and Rococo designers, such as Dominikus Zimmermann, Johann Michael Fischer, and Jan Krzysztof Glaubitz. All façades of buildings, both public and sacred, designed by D. Zimmerman were painted to emphasize vertical articulation, e.g., the highly ornamental front of the town hall in Landsberg am Lech (1719), the pilgrimage church in Steinhausen (1729), or the more modest parish church in Buxheim (1729). Among Fischer's works with such an order painting using a two-color scheme of grey and white are the tower façades of the Church of the Holy Sepulchre in Deggendorf (1722–1727) and the monastery church of Marienmünster Dießen in Dießen am Ammersee (1731–1739) or, this time with a strongly contrasting ochre detail on a white background, the monastery church of St. Michael in Berg am Laim (1738–1751). In J.K. Glaubitz's designs, the continuity of multiplied elements of vertical division was emphasized by color in almost all façades of the Vilnius churches, e.g., the Church of St. John the Baptist and St. John the Evangelist (1738–1749, Figure 17a), St. Catherine (1741–1744), or Holy Spirit Church (1753–1770). Similar solutions were applied by Antonio Paracca, who worked at the same time in Vilnius on the Basilian Monastery Gate, and the façade of the Missionary Church (1751–1756) (Karpowicz 2011), as well as by the Jesuit Paweł Giżycki, author of the façades of the Jesuit church in Krzemieniec (1731–1745), Stanislavov (Ivano-Frankivsk, 1752–1763), and the Bernardine Monastery in Lutsk (1752–1789) (Betlej 2000a, 2000b). In Cracow, this polychrome effect emphasizing verticality first appeared on church façades by Antonio Solari, who redesigned the façade of the church of St. Michael the Archangel and St. Stanislaus on Skałka (1733–1749) (Lenartowicz 2003) (restored in 2019). In addition, on the front façade of the Piarist Church of the Transfiguration of Our Lord (1759–1761, Figure 17b) by Franciszek Placidi (restored in 2007) (Polskie Pracownie Konserwacji Zabytków S.A 2012), similar in expression to the examples from Vilnius, the painting of the entablature was done in this way.

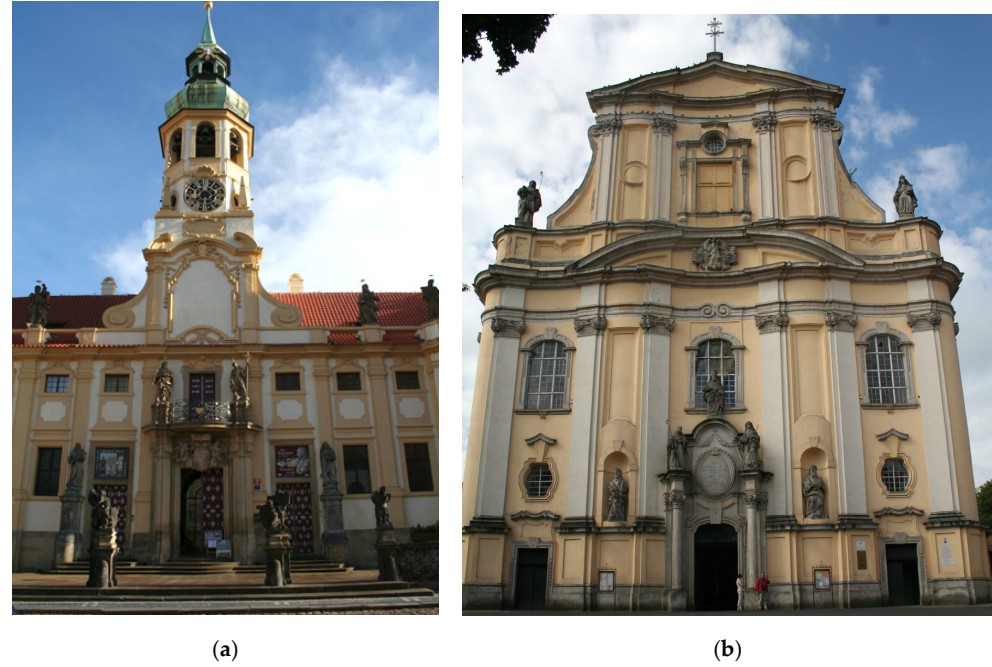

(**a**)  (**b**)

**Figure 16.** Continuity of multiplied elements of vertical division emphasized by color: (**a**) the monumental façade of the Loreto complex in Prague by Ch. and K.I. Dienzenhofer (1722–1726); (**b**) the façade of monastery church in Lubomierz (1728–1730, builder J.J. Scheerhofer). Photos by author.

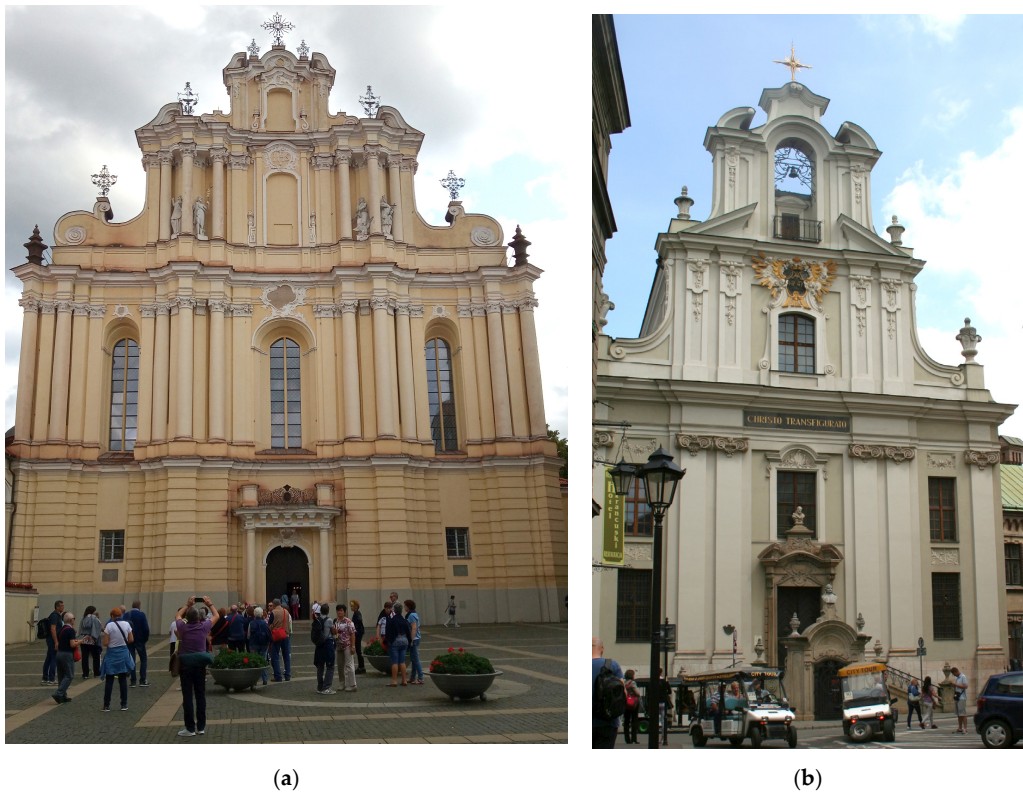

(**a**)  (**b**)

**Figure 17.** The late Baroque examples of highlighting the vertical articulation of the architectural structure with color: (**a**) the façade of the church of St. John the Baptist and St. John the Evangelist in Vilnius (1738–1749); (**b**) the façade of the Piarist Church in Cracow (1759–1761). Photos by author.

Further examples of such a system of polychrome façade order can be found among the continuations and imitations of South German and Czech models, such as the façades of the Church of the Purification of the Blessed Virgin Mary in Dub nad Moravou (1756), the Minorite Church of St. Anthony in Eger (1758–1773) (recently repainted from red to ochre in 2013), and the works of the next generation of artists from the Como region working in Italy, such as Andrea Nono, author of the Rococo façades of the Church of the Holy Trinity in Crema (1740). Tuscan Bartolomeo Rastrelli, working in Ukraine and Russia, designed the façades of St. Andrew's Cathedral in Kiev (1747 and 1754), the Smolny Monastery in St. Petersburg (1748–1764, Figure 18), and the Winter Palace (1754–1762) in the same color scheme.

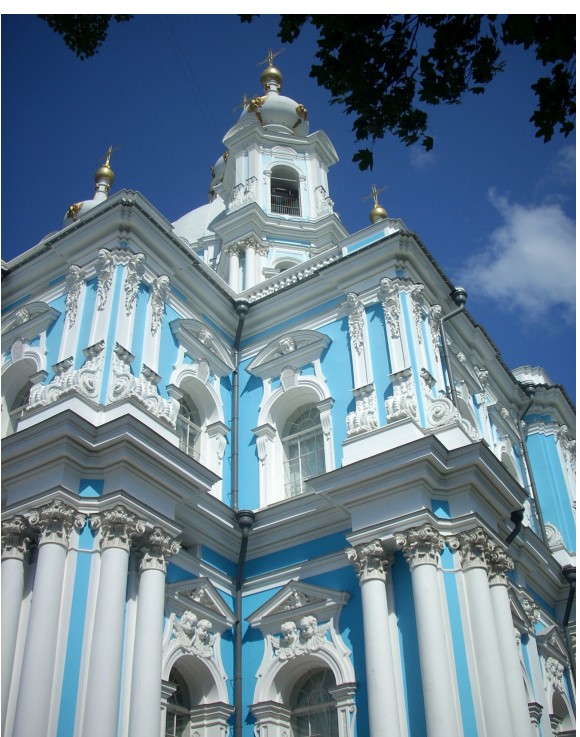

**Figure 18.** Despite the ahistorical color, the polychromatic distinction of the vertical structure was preserved. The Smolny Cathedral in St. Petersburg (1748–1764). Photo from private collection.

## 5. Discussion: The Role of Color Arrangement of the Components of Architectural Order in Shaping Compositional Expression

In the Baroque, the vertical direction of the composition was strongly emphasized by multiplying or applying perspective arrangements of supports, and finally by embattling cornices (Figure 19a). However, in the system of façade tectonics, the moment of stopping the upward flight of the eyes in the form of horizontal massive strips of crowning entablature remained. This weakened the dynamics of the structure, but emphasized its monumentality. A change was brought about only by decisions regarding the material and color separation of elements of the frieze above the supports within the embattled entablature as well as the type of imposts above the chapiters of semi-columns or, more frequently, pilasters (Figure 19b). The uniform color of all vertical elements of the façade structure, sometimes interrupted only by a small, very decorative fragment of differently treated capitals, guaranteed an unambiguous verticality of the composition. It is characteristic that from the very beginning, the introduction of successive stages of this solution, from molding of the cornice to color separation of the pseudo-imposts, was connected with the influence of the medieval tradition, e.g., with the designs of the Milan Cathedral façade or with the introduction of the concept of post-Reformation church façade in the countries where, until then, the Gothic forms dominated in architecture. It also depended on the

personal inspiration of the designers in strongly expressive Gothic structures, as in the case of the artists of the mature and late Baroque of Central Europe. These decisions on the disposition of colors within the entablature in the architectural order had a particularly significant influence on the final expression of the work.

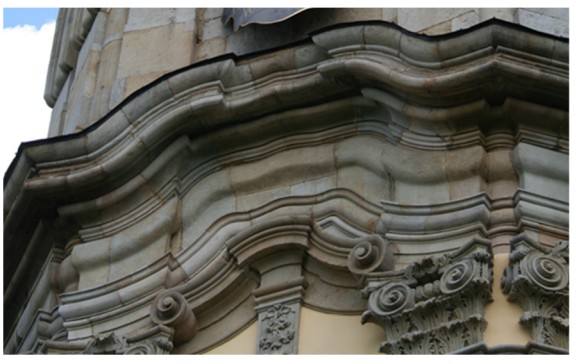 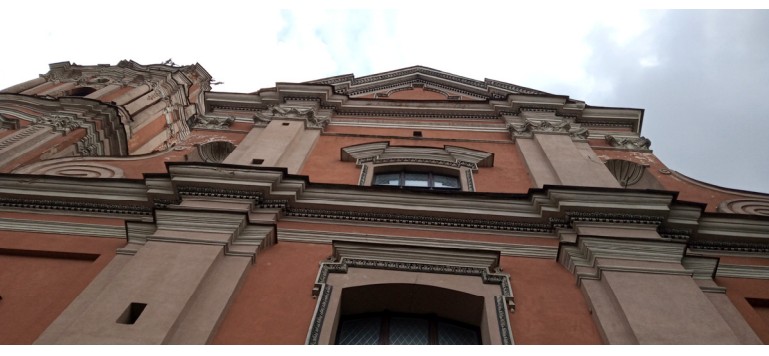

(**a**)                                          (**b**)

**Figure 19.** The Baroque entablature: (**a**) the late Baroque embattling entablature from the monastery church in Krzeszów; (**b**) the material and color separation of elements of the frieze above the supports in the entablature of early the Baroque church of All Saints in Vilnus (ca. 1660). Photos by author.

In most cases, the question of whether the color solution was developed by the architect or one of the contractors remains unresolved. However, even if it was the choice of one of the executors, in the case of Baroque buildings, a thorough understanding of the architectural expression of the façade structures is repeatedly demonstrated. It is interesting that this applies not only to the works of the most eminent artists, but also to more modest buildings from more provincial areas. Perhaps this was due to a very conscious copying of patterns and an understanding of the artistic intent of the designers.

## 6. Conclusions: Significance of Conservators' Decisions in the Field of Coloring and Shaping the Compositional Effect of Baroque Architecture

It is uncertain whether all solutions of color schemes originate from the times when the buildings were constructed or resulted from careful stratigraphic research conducted by conservators. Some polychromes may have been lost permanently; sometimes the later layers are wrongly interpreted as the original ones, which is reflected in decisions on radical changes of the architectural color arrangement restored after the latest research. In recent times, great emphasis was put on correct identification of original polychrome. It seems, however, that in some cases it is limited to determining the colors and pigments used, and not their exact distribution on the façade. Often, the upper parts of the building are heavily damaged, which makes it difficult to decipher correctly the coloring of the entablature, and in particular, to resolve the issue of the color split of the frieze field.

As can be seen from the presented considerations (Philippot [1988] 1998), the meaning of the impact of an architectural work in the surrounding space is significantly influenced by the choice of color. However not everyone is aware the importance, in expression of baroque façades, of the decisions concerning the color continuity of the frieze or of the introduction of pseudo-imposts distinguished in color above the supports. This can be illustrated by examples, where the vertical ordering of the architectural structure is changed to an arrangement with an accentuated entablature by the uniform color of the frieze; without the possibility of research, in the case of reconstructed buildings, it is decided to introduce a pattern with colorful imposts (the Jesuit church in Warsaw, where in 2008 a polychrome arrangement was designed, referring to analogous color solutions, probably without realizing that it would be one of the earliest examples of this solution) (Poplatek and Paszenda 1972; Paszenda 2010).

It is inevitable that design decisions concerning the restoration of the color system of baroque façades are based on conservation research, which cannot answer all the questions involved due to certain limitations. It is necessary to use general knowledge as well as information drawn from analogous projects or those created in a similar circle of artists, place or time. This article can contribute to such considerations in the field of identifying the color scheme of architectural orders.

**Funding:** This research received no external funding.

**Data Availability Statement:** The study did not report any data.

**Acknowledgments:** Photos in Figure 1a,b, Figures 3b, 4, 5b, 12 and 13 by Sebastian Wróblewski and in Figure 18 by Marta Galantowicz, with agreement for publication in this article.

**Conflicts of Interest:** The author declares no conflict of interest.

## Appendix A

A1: J.M. Bernardoni had already designed the stone façades of the churches of San Michele in Cagliari (1578) and Santa Caterina in Sassari (1579–1609), and in such a way, composed similar arrangements for Jesuits in Jarosław, Nieśwież (1586–1593), and Kalisz (before 1586) (Betlej 2018).

A2: Carlo Antonio Carlone (1635–1708), a native of Val'Intelvi, in his reconstructions of the façades of large monasteries and monastic, pilgrim, and parish church façades in Upper Austria in Schlierbach (1680–1683); continued by Jacob Prandtauer in Sankt Florian (1686–1705), Christkindl, Kremsmünster, and Garsten, who introduced an expressive vertical composition.

A3: The stone (Eggenburger Kalksandstein) elements of the architectural order on the façade of the imperial residence were plastered and painted in shades of white and set off against an orange-red brick background (realized by 1711; the façades were repainted the characteristic yellow in the 19th century).

A4: Briano duplicated the pattern of embattled entablature from the Northern Italian solution, also used in the church in Palermo, in the construction of which he participated. However, perhaps due to material shortages, he ordered only reinforcements in the frieze part over the supports of stone.

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
