# Peer review of "The Polychrome in Expression of Baroque Façade Architecture"

_arts, 1984_

Round 1

Reviewer 1 Report

The paper deals with a theme that is not altogether original but certainly particularly interesting and neglected by studies.

The research is conducted methodically, the bibliographic references are sufficient but, as the author says, the cromatic data are not always reliable and readily available only in work relationships and rarely in the form of publications as research results.                       

There are numerous references to ecclesiastical buildings in the text but only some are illustrated with photographs.

It is necessary to introduce references to illustrations in the text for a more comfortable reading.

Graphic illustrations are necessary to summarize and schematize the dispotition of the different chromatic solutions used and what role they played in contributing to the definition of the Baroque facade.

Author Response

I would like to thank the Reviewer for the high rating of my work. 

I have tried to make corrections as recommended. 

I added references in the text to the illustrations.

I have supplemented the article with new illustrations. I checked and completed the names of all the others architectural objects to allow flawlessly find photographs of them in Internet resources.

In accordance with the Reviewer's explicit recommendation, I have introduced illustrations into the chapter summarizing the information.

Reviewer 2 Report

Interesting and well-prepared article.

Author Response

I am grateful for such a high rating of my text. I am glad that it met with the interest and recognition of the Reviewer.